



# From atmospheric water isotopes measurement to firn core interpretation in Adelie Land: A case study for isotope-enabled atmospheric models in Antarctica

Christophe Leroy-Dos Santos[1,2], Elise Fourré[1], Cécile Agosta[1], Mathieu Casado[1], Alexandre Cauquoin[3], Martin Werner[4], Benedicte Minster[1], Frédéric Prié[1], Olivier Jossoud[1], Leila Petit[1], and Amaelle Landais[1]

[1]Laboratoire des Sciences du Climat et de l'Environnement, LSCE-IPSL, CEA-CNRS-UVSQ, Université Paris-Saclay, Gif-sur-Yvette, France
[2]Centre for Environmental and Marine Studies (CESAM), Department of Physics, University of Aveiro, Portugal
[3]Institute of Industrial Science (IIS), The University of Tokyo, Kashiwa, Japan
[4]Alfred Wegener Institute (AWI), Helmholtz Centre for Polar and Marine Research, Bremerhaven, Germany

**Correspondence:** Christophe Leroy-Dos Santos (christophe.leroy@lsce.ipsl.fr)

**Abstract.**

In a context of global warming and sea level rise acceleration, it is key to estimate the evolution of the atmospheric hydrological cycle and temperature in the polar regions, which directly influence the surface mass balance of the Arctic and Antarctic ice sheets. Direct observations are available from satellite data for the last 40 years and a few weather data since the 50's in

Antarctica. One of the best ways to access longer records is to use climate proxies in firn or ice cores. The water isotopic composition in these cores is widely used to reconstruct past temperature variations.

In order to progress in our understanding of the influence of the atmospheric hydrological cycle on the water isotopic composition, we first present a 2-year long time series of vapor and precipitation isotopic composition measurement at Dumont d'Urville station, in Adélie Land. We characterize diurnal variations of meteorological parameters (temperature, humidity and

10 $\delta^{18}O$) for the different seasons and determine the evolution of key relationships ($\delta^{18}O$ versus temperature or humidity) along the year: we found mean annual slopes of 0.5 and 0.4 ‰ °C$^{-1}$ for the relationship of $\delta^{18}O$ vs. temperature in the water vapor and in the precipitation respectively. Then, this data set is used to evaluate the Atmospheric General Circulation Model ECHAM6-wiso (model version with embedded water stable isotopes) in a region where local conditions are controlled by strong katabatic winds which directly impact the isotopic signal. We show that a combination of continental (79%) and oceanic

(21%) grid cells leads model outputs (temperature, humidity and $\delta^{18}O$) to nicely fit the observations, even winter extreme synoptic events are represented in the model. Therefore we demonstrate the added value of long-term water vapor isotopic composition records. Then, as a clear link is found between water vapor and precipitation isotopic composition, we evaluate how isotopic enabled models can help interpreting short firn cores.



## 1 Introduction

East Antarctica is the biggest fresh water reservoir on Earth (Smith and Evans, 2007). In a context of global warming, it is key
to monitor and anticipate the surface mass balance (SMB) of this region and its link with climate change. Adélie Land is part of
the Wilkes Land Coast, a region that is at the boundary between the eastern part of the Antarctic plateau and the Indian ocean.
Recent studies based on remote observations, reanalysis or CMIP5 models disagree on the recent evolution of temperature in
this region (Wang et al., 2020; Retamales-Muñoz et al., 2019; Stenni et al., 2017), showing that it is complicated to determine
how it is impacted by global warming.

Antarctica climate and ice sheet altitude evolution have been measured for four decades based on remote sensing and show
an increasing mass loss in Wilkes Land (Rignot et al., 2019). These reconstructions require calibrations or evaluation with
ground based measurements but weather stations and direct observations of surface mass balance are sparse (Favier et al.,
2013; Wang et al., 2021). Global reanalysis data are suitable for studying climate variability, but these products have only been
reliable in the data-scarce Antarctic region since 1979, when they began assimilating satellite data (Marshall et al., 2022). For
periods older than 40 years, Antarctic climate reconstructions rely on a few weather stations installed in the 50's (Fogt et al.,
2016) or on the interpretation of climate proxies. Water isotope measurements in firn and ice cores are key tools to provide
reconstruction of past climate and atmospheric water cycle over the last centuries and millennia (Jouzel et al., 2013; Stenni
et al., 2017). They have been extensively used for providing past temperature reconstructions in continental Antarctica where
water isotopic depletion can be related to temperature decrease through increased distillation along the atmospheric moisture
path from evaporative regions to the precipitation sites of interest (Jouzel et al., 2007; Stenni et al., 2011). The interpretation
of water isotope records in coastal Antarctica is more difficult because distillation is not the sole dominant influence on the
water isotopic signal. Local effects of ocean evaporation itself influenced by the presence of sea ice, katabatic wind direction
and speed, remobilisation or sublimation of surface snow may also have a strong effect on the water isotopic composition of
the deposited snow and hence on the archived signal (Ekaykin et al., 2002; Casado et al., 2018). While these effects prevent a
simple interpretation of water isotopes as temperature proxy, they open the way to the use of these tracers to better constrain
the past atmospheric water cycle.

In fact, recent studies have shown that isotopic signals recorded in coastal ice or firn cores are poorly correlated to surface
temperature (Goursaud et al., 2017, 2019) or to SMB (Schlosser et al., 2014). This is confirmed by Altnau et al. (2015) who
observed weaker relationship between SMB and $\delta^{18}$O for coastal firn cores in comparison to inland drilling sites with a 76 firn
core dataset in Dronning Maud Land. Finally, these studies point out the need to take into account the influences of atmospheric
dynamics and local processes, in addition to the classical thermodynamics part on isotopic signals. For instance, Sinclair
et al. (2014) studied sea ice variations in the Ross Sea using d-excess signal, a second order parameter reflecting evaporative
conditions, coupled with chemical indicators in a core from Whitehall Glacier. This study attributes the observed variations to
enhanced southerly winds and an increased advection of sea ice to the north showing regional dynamics influences. Also, from
shallow firn cores drilled in Fimbul Ice Shelf in western Dronning Maud Land, Vega et al. (2016) suggest that d-excess record
reflects the effect of seasonal moisture transport changes.



Furthermore, other ice core interpretation techniques, such as virtual firn cores (Sime et al., 2011; Casado et al., 2020) that take into account the intermittency of precipitation and isotope diffusion in firn on isotopic signal, encourage the use of isotope-enabled models in order to consider other processes than temperature that affect the primary isotopic signal (Sime et al., 2011; Casado et al., 2020). For example, Goursaud et al. (2018) demonstrated how ECHAM5-wiso could be valuable to investigate the dynamic of water stable isotope composition in precipitation in regard to different locations or seasons.

Here we build on these previous studies and aim at better constraining the isotopic signal of water vapor and precipitation in coastal sites to improve the interpretation of firn and ice core archives. This can be achieved by measuring the isotopic composition of precipitation and water vapor on site in order to understand local dynamics influences and using atmospheric general circulation models (AGCMs) equipped with water vapor isotopes outputs that already includes various processes linked to atmospheric dynamics. For example, large diurnal cycles observed in isotopic composition of water vapor at Dome C were associated with atmospheric regimes demonstrating impact of local meteorological processes on isotopic signal (Casado et al., 2016). At Kohnen, Ritter et al. (2016) showed that amplitude of isotopic diurnal variations were significantly lower in isotope-enabled AGCMs (ECHAM5-wiso and LMDZ5Aiso) than in observations. In coastal regions, it has been observed that the isotopic signature of maritime air mass contrasts with depleted cold glacial air mass at Syowa station (Kurita et al., 2016) while in Adélie Land, a first summer campaign of measurements made it possible to understand the influence of katabatic wind dynamics on the diurnal isotopic cycle (Bréant et al., 2019). Also, a two-year data series at Neumayer station III was used to compare isotopic signal in water vapour to back trajectory simulations (Bagheri Dastgerdi et al., 2020).

Here we present a long-term study of continuous isotopic measurement of water vapor and precipitation at Dumont d'Urville (DDU). First, we present our instrumental set-up and the analysis of a 2-year isotopic series observed at DDU. Next we look in detail at ECHAM6-wiso outputs at DDU geographical position and evaluate the model performance. Finally, we discuss the added value of an isotope-enabled model evaluated for ice-core interpretation in coastal regions by comparing a ECHAM6-wiso based virtual firn core to a firn core drilled close to DDU and presented in a previous study (Goursaud et al., 2017).

## 2 Materials and Methods

### 2.1 Observations

#### 2.1.1 Water vapor isotopic composition

A Picarro analyzer (L2130-i) was installed in December 2018 in Adélie Land, at the coastal DDU station (Fig. 1) to monitor the isotopic composition of atmospheric water vapor. We used the same configuration as in Bréant et al. (2019), i.e. an inlet is positioned 1 meter above the shelter roof, 6 meters above ground level, roughly 30 meters above sea level (a.s.l.). This inlet is connected to the isotopic analyzer by a heated and insulated perfluoroalkoxy (PFA) tube. The inlet is covered by a Gelman Zefluor 0.5 $\mu m$ filter to prevent any inflow of precipitation, blowing snow or penguins' feathers. Continuous measurement of the isotopic composition of atmospheric water vapor is still running to date. Here we present the first two years of measurement, from 1st January 2019 to 31st December 2020.



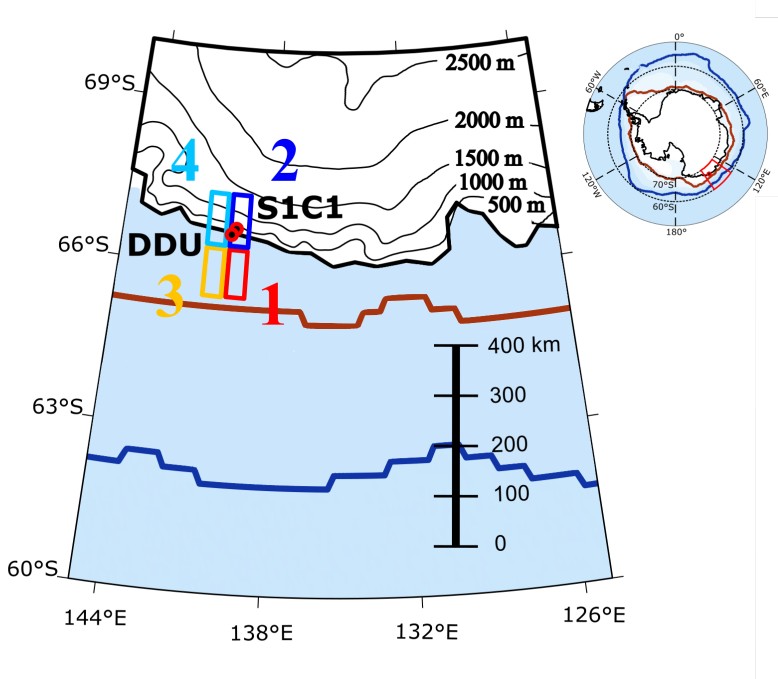

**Figure 1.** Adelie land map with Dumont d'Urville scientific station (located on Petrel Island, 5 km from the coast) and S1C1 drilling site (10 km inland) indicated by red dots. Black contours represent the topography, bold red and blue lines represent the sea-ice extent, respectively in March and October, defined as areas with a minimum sea-ice fraction of 0.15 from the ERA5 reanalysis over the period 1979-2020, as in Ding et al. (2017). Labelled (from #1 to #4) and colored (in red, blue, orange and light blue) rectangles represents closest ECHAM6-wiso grid cells to DDU.

In the following study we discuss molecular water vapor mixing ratio (in ppmv), hereafter called humidity, and water vapor $\delta^{18}$O and $\delta$D measured by the laser spectrometer. The raw datasets have been calibrated and corrected following the protocol outlined in previous studies (Tremoy et al., 2011; Bonne et al., 2014; Steen-Larsen et al., 2014) and described below.

Humidity values have been compared to independent measurements from Meteo France weather station at DDU (Fig. S1 in Supplement). The comparison shows a robust linear relationship between both measurements over the 2-year period (coefficient
of determination, hereafter called $R^2$, equal to 0.99 and slope of 0.98). In the following, we only display raw humidity data from Picarro without any correction.

The calibration of the $\delta^{18}$O and $\delta$D series was performed following the approach described in Leroy-Dos Santos et al. (2020). Three main corrections were applied: 1) the influence of humidity on $\delta^{18}$O and $\delta$D measurement, 2) the difference between the measured values and the true isotopic values, and 3) the temporal drift of the instrument. For the calibration procedure, we used
both a specifically designed low humidity level generator described in Leroy-Dos Santos et al. (2021) and a standard delivery



module (SDM) associated with a Picarro brand vaporizer for higher humidity values. For consistency with the international VSMOW-SLAP scale (International Atomic Energy Agency, 2006), we relied on two bracketing internal standard waters: NEEM ($\delta^{18}$O: $-33.56 \pm 0.05‰$, $\delta$D: $-257.6 \pm 0.5‰$) and FP5 ($\delta^{18}$O: $-50.64 \pm 0.05‰$, $\delta$D: $-395.9 \pm 0.5‰$), calibrated at LSCE with mass spectrometry for $\delta^{18}$O and with laser spectrometry for $\delta$D.

To determine the isotope-humidity calibration, vapor with known isotopic composition was generated at different humidity levels, from 150 to 1500 $\mathrm{ppmv}$ with the low humidity level generator (in the field) and from 1000 to 5500 $\mathrm{ppmv}$ with the SDM (at LSCE). A well constrained relationship is determined from the 2018 data set over the whole range of humidity values (Fig. S2). Further measurements in 2019 and 2020 show that this calibration is stable over time, as already noted by Bailey et al. (2015). This calibration curve also takes into account the shift between measured and true isotopic values. The same shift

in $\delta^{18}$O and $\delta$D has been observed between measured and true value for both NEEM and FP5. We checked the drift of the instrument by measuring every 48-hour NEEM standard at 1100 $\mathrm{ppmv}$ using an automatic routine. Some technical issues led us to select only 150 calibration over the 2 years period. The correction associated with the mean linear drift is insignificant with respect to the humidity dependency correction and we estimate the mean uncertainty as 0.8 ‰ and 3.2 ‰ for $\delta^{18}$O and $\delta$D respectively.

**2.1.2    Precipitation isotopic composition**

In parallel with continuous water vapor isotopic measurements, the isotopic composition of precipitation and surface snow is also monitored. Precipitation is collected on a daily basis whenever the amount of precipitation is sufficient using a wooden platform with a plastic bottom (length x width x height = 60x40x10 $\mathrm{cm}$) installed at DDU station on building rooftop. Samples are sent back in -20°C shipment to LSCE once a year. Measurements are performed with a L2130-i Picarro laser spectrometer

working in liquid mode. The uncertainty (1 sigma) of our data set is 0.05 ‰ and 0.2 ‰ respectively for $\delta^{18}$O and $\delta$D. It is estimated using replicates over 15 % of the samples.

**2.1.3    Meteorological data**

Meteorological data are available since 1956 at the Meteo France weather station of DDU at 3-hour resolution and 1-hour resolution. Hereafter, we use the 2-meter air temperature (°C), the specific humidity (volume mixing ratio in $\mathrm{ppmv}$), calculated

from pressure ($\mathrm{mbar}$), temperature and relative humidity (%), as well as the wind speed ($\mathrm{m\,s^{-1}}$) and direction (°).

**2.2    Models**

**2.2.1    ERA5 reanalysis**

We use outputs from the global atmospheric reanalysis of the European Center for Medium-Range Weather Forecasts (ECMWF), ERA5 at hourly resolution data from 2019 to 2020: the total daily precipitation amount ($\mathrm{kg\,m^{-2}\,day^{-1}}$), the sea ice area frac-

tion, the specific humidity ($\mathrm{kg\,kg^{-1}}$), the surface pressure ($\mathrm{Pa}$) and the 2m-temperature (°C) (Hersbach et al., 2020).





### 2.2.2 ECHAM6-wiso

ECHAM6-wiso is the isotopic version of the atmospheric general circulation model ECHAM6 (Stevens et al., 2013). The implementation of the water isotopes in ECHAM6 has been described in detail by Cauquoin et al. (2019), and has been updated in several aspects by Cauquoin and Werner (2021) to make the model results more consistent with the last findings

based on water isotope observations (isotopic composition of snow on sea ice taken into account, supersaturation equation slightly updated, and kinetic fractionation factors for oceanic evaporation assumed to be independent of wind speed). We have used ECHAM6-wiso model outputs from a simulation at high spatial resolution (at T127L95, 0.9° horizontal resolution and 95 vertical levels) nudged to ERA5 reanalysis. The ECHAM6-wiso 3D-fields of temperature, vorticity and divergence as well as the surface pressure field were nudged toward the reanalysis data every 6 hours. The orbital parameters and greenhouse gases

concentrations have been set to the values of the corresponding model year. The monthly mean sea surface temperature (SST) and sea-ice fields from the ERA5 reanalysis have been applied as ocean surface boundary conditions, as well as a mean $\delta^{18}O$ of surface seawater reconstruction from the global gridded data set of LeGrande and Schmidt (2006). As no equivalent data set of the $\delta D$ composition of seawater exists, the deuterium isotopic composition of the seawater in any grid cell has been set equal to the related $\delta^{18}O$ composition, multiplied by a factor of 8, in accordance with the observed relation for meteoric water

on a global scale (Craig, 1961). The ECHAM6-wiso simulation is described in detail and evaluated in Cauquoin and Werner (2021).

## 3    Results

### 3.1    Vapor and precipitation records

We present the full 2019-2020 record of hourly atmospheric vapor isotopic composition at DDU (Fig. 2). The mean temperature

and humidity over the two-year measurement period (-11.2 °C and 1883 ppmv) are close to the average value calculated over the full dataset available from Meteo France since 1957 (-10.8 °C and 1826 ppmv), and show similar seasonal cycles (Fig. 2). A clear seasonal cycle is observed for all variables except d-excess (Tab. 1), with higher mean values in summer (-2.3 °C, 3354 ppmv and -27.5 ‰ for temperature, humidity and $\delta^{18}O$ respectively in December, January and February (DJF)) than in winter (-17.0 °C, 1185 ppmv, -34.8 ‰ in June, July and August (JJA)).

The variability of the measured variables (temperature, $\delta^{18}O$, d-excess and humidity) also shows a seasonality (Tab. 1), with higher standard deviation in winter than in summer. These results can be explained by a different impact of the synoptic events, which transport warm and moist air toward DDU and correspond to rapid increases of humidity and temperature (Fig. 2). Those events, seen as meridional exchanges, leads to higher variability in winter while there is a larger meridional temperature and moisture gradients than in summer. Blue bars in Figure 2 show the distribution of the main synoptic events associated

with daily precipitation amounts higher than $4.5 \, \mathrm{kg \, m^{-2} \, day^{-1}}$. We detect 35 precipitation peaks over the 2-year period, 7 (20 %) during DJF (summer) and 8 (23 %) during JJA (winter). Among the different synoptic events in winter, two major events are particularly intense during extended winter (from May to September): (a) 23 July 2019, with a precipitation rate of 31



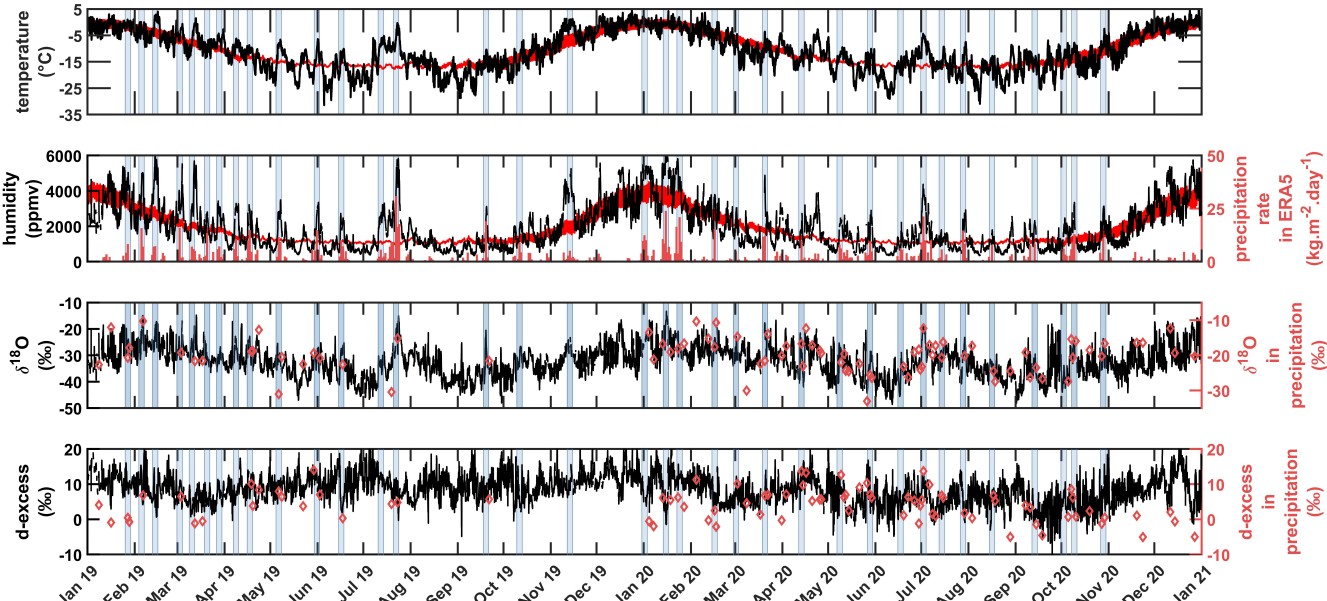

**Figure 2.** Two-year data series (1 Jan. 2019 to 31 Dec. 2020) of meteorological and isotopic measurements at DDU. Panels from top to bottom: 1) hourly 2m-temperature (°C) from Meteo France weather station in black, 3-hour temperature averaged over 1957-2020 period in red. 2) hourly humidity (ppmv) measured by the Picarro laser spectrometer in black, mean humidity (ppmv) from Meteo France weather station averaged over the 1957-2020 period in red; red bars indicate daily precipitation rate from ERA5 reanalysis (in $\mathrm{kg\,m^{-2}\,day^{-1}}$). 3) calibrated $\delta^{18}$O (‰) in water vapor (hourly average in black) and in precipitation (red diamonds). 4) d-excess (‰) in water vapor (hourly average in black) and in precipitation (red diamonds). Light blue bars indicate high daily precipitation rates (peaks) with value higher than 4.5 $\mathrm{kg\,m^{-2}\,day^{-1}}$ and separated by a minimum of 5 days.

$\mathrm{kg\,m^{-2}\,day^{-1}}$ and (b) 2 July 2020, with a precipitation rate of 21 $\mathrm{kg\,m^{-2}\,day^{-1}}$. These events correspond to the largest daily precipitation rates of each winter, and to the first and third largest daily precipitation amounts when considering the whole
2019-2020 period. The values of temperature, humidity and $\delta^{18}$O during these events (-1 °C and -4.4 °C, 5780 ppmv and 4370 ppmv, -17.8 ‰ and -19.3 ‰, respectively for the two events) are close or above summer averages (Tab. 1).

Differences between winter and summer weather regimes impact the relationship between variables. First, humidity and $\delta^{18}$O show high correlation coefficients (calculated from daily means) both over the whole record ($R^2$=0.6) and on a seasonal scale ($R^2$=0.5 for DJF and JJA). The slope of this linear relationship (Tab. 1 and Fig. S3) is almost doubled during winter
($4.5.10^{-3}$ ‰ $\mathrm{ppmv}^{-1}$) compared to summer ($2.4.10^{-3}$ ‰ $\mathrm{ppmv}^{-1}$). Second, the linear relationship between $\delta^{18}$O and temperature is strong for the full period ($R^2$=0.5) but vanishes during summer ($R^2$=0.1). This can be related to the smaller daily variability during summer. The $\delta^{18}$O-temperature slope (Tab. 1 and Fig. S4) over the full period (0.5 ‰ $°C^{-1}$) is different from the winter mean slope (0.6 ‰ $°C^{-1}$, $R^2$=0.4).





**Table 1.** Mean and standard deviation (std) of temperature (temp.), humidity (hum.) and isotopic composition calculated from daily means (left). Correlations are calculated with daily data (right). The period considered is 2019-2020 except for "historical" (hist.) data which are the mean of daily averages over 1957-2020.

| | Temp. (°C) | | Hum. (ppmv) | | | $\delta^{18}O$ (‰) | | d-excess (‰) | | $\delta^{18}O$ vs Hum | | $\delta^{18}O$ vs Temp. | |
|---|---|---|---|---|---|---|---|---|---|---|---|---|---|
| | mean | std | mean | std | relative std (%) | mean | std | mean | std | slope (‰ $ppmv^{-1}$) | $R^2$ | slope (‰ $°C^{-1}$) | $R^2$ |
| DJF | -2.3 | 2.4 | 3373 | 998 | 30 | -27.4 | 3.5 | 10.2 | 3.2 | $2.4\ 10^{-3}$ | 0.45 | // | 0.06 |
| JJA | -16.9 | 5.0 | 1194 | 783 | 66 | -34.7 | 5.0 | 7.8 | 3.3 | $4.5\ 10^{-3}$ | 0.51 | 0.64 | 0.42 |
| Total | -11.2 | 7.1 | 1833 | 1196 | 65 | -32.3 | 5.2 | 8.4 | 3.4 | $3.4\ 10^{-3}$ | 0.61 | 0.52 | 0.51 |
| Hist. | -10.8 | 5.9 | 1826 | 902 | 49 | // | // | // | // | // | // | // | // |

As mentioned above, synoptic events are not clearly visible in summer. Summer variability is actually dominated by the
170 succession of diurnal cycles. In Figure 3, we show the mean diurnal cycles in summer and winter. During winter, the diurnal
cycles of temperature and humidity are flattened to 0.6 °C and 40 ppmv, and are not visible for $\delta^{18}O$ and d-excess (Fig. 3 and
Tab. 1). The summer diurnal cycle amplitudes reach almost 4°C, 1000 ppmv and 4 ‰ for temperature, humidity and $\delta^{18}O$,
respectively. The d-excess summer diurnal cycle amplitude (about 1 ‰) is smaller than the uncertainty and therefore hardly
significant. The summer amplitudes of temperature and humidity diurnal cycles over the 2019-2020 study period are similar
to the diurnal variability in temperature and humidity over the whole instrumental period (1957-2020). This summer diurnal
cycle has been documented in previous detailed studies (Pettré et al., 1993; Bréant et al., 2019) and attributed to katabatic wind
variability particularly during clear sky conditions, when the sun zenithal angle impacts the radiative cooling of the continental
surface responsible for the katabatic flow. When the sun is at its lowest position, cold and dry air masses coming from inland
are associated with low $\delta^{18}O$ values. On the contrary, when the sun is at its highest position, the origin of atmospheric air is
180 more local, through convective mixing for instance, and we observe a parallel increase in humidity, temperature and $\delta^{18}O$.

Over the 2019-2020 period, we collected 82 precipitation samples (Fig. 2). The condensation of vapor in the upper atmo-
spheric layers leads to precipitation but subsequent exchanges between atmospheric water vapor and snow flakes can also affect
the isotopic composition of the collected precipitation. In order to compare isotopic signals in precipitation and vapor, we cal-
culate the theoretical isotopic composition of vapor at equilibrium with each precipitation sample. We use solid-vapor fraction-
185 ation coefficients at equilibrium from Majoube (1971) and Merlivat and Nief (1967) calculated with the daily 2m-temperature
corresponding to the day of sample collection. The comparison is made with the daily averaged isotopic composition of water
vapor measured at DDU (Fig. S5). Results exhibit a clear linear relationship between both datasets ($R^2$= 0.5) with a slope
of 0.7 ‰ $‰^{-1}$. Note that we do not expect a correlation coefficient and slope of 1 because we averaged over 24h while the
precipitation event is often shorter and precipitation samples can be affected by post-deposition effects before their collection.



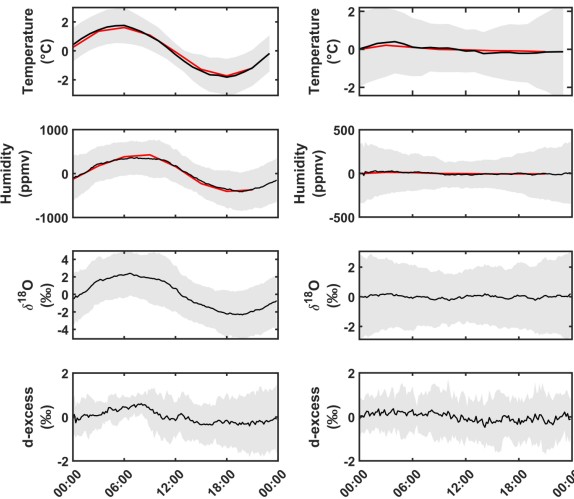

**Figure 3.** Mean diurnal cycles estimated from hourly data (UTC+10) in summer (DJF, left panel) and winter (JJA, right panel) over the 2019-2020 period. Red lines are computed from the historical Meteo France weather station data set (1957-2020 period, 3-hourly resolution).

## 3.2 ECHAM6-wiso: data-model comparison at DDU

We present the results of the comparison between ECHAM6-wiso outputs and DDU measurements of temperature, humidity and $\delta^{18}$O. We consider the four ECHAM6-wiso grid cells around DDU station (Fig. 1). Since isotopic variables are not available at 2 m above the ground in ECHAM6-wiso, we consider all modeled variables at the lowest atmospheric level in ECHAM6-wiso (approximately 75 m above the ground, see exact altitudes in Tab. S1) in order to work with an homogeneous data set. We have combined oceanic (continental) cells using the average values of grid cells outputs #1 and #3 (#2 and #4) to plot Figure 4. Indeed, both oceanic (continental) cells show very similar characteristics (i.e. mean values, standard deviation, correlations with DDU measurements; Fig. S6 and Tab. S1).

Modeled temperature always present a cold bias (-1.2 °C for the oceanic grid cell and -6.7 °C for the continental grid cell) compared to measurements at DDU, while humidity bias is either positive (about 500 ppmv in average for oceanic grid cells) or negative (about -600 ppmv in average for continental grid cells). The temperature bias is not related to the average mean altitude of grid cells, as a similar comparison using the modeled 2m-temperature also results in cold biases (Tab. S1). We observe a bias of -3 °C in ERA5 2m-temperature temperature at DDU in comparison to measurements over the 2 years data set which could explain the differences in temperature between model and measurements as suggested by Goursaud et al. (2018). Despite these biases, ECHAM6-wiso outputs reproduce well the variability of the temperature and humidity records at DDU for both oceanic and continental grid cells. Over the full period, modeled daily temperature and humidity are highly correlated to observations ($R^2 = 0.9$, Tab. S1) and both show very similar standard deviations (about 8 °C and 1200 ppmv for continental or oceanic cells, respectively for temperature and humidity, compared to 7 °C and 1200 ppmv for observations, Tab. 1).





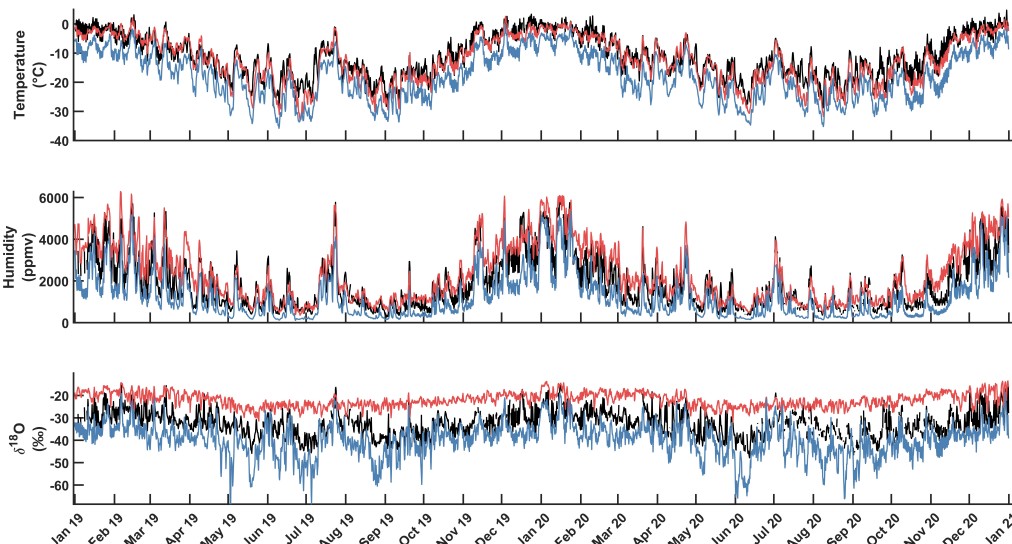

**Figure 4.** Meteorological and isotopic measurements at DDU at 6-hourly resolution, in black. Panels from top to bottom: 1) 2m-temperature (°C) from Meteo France weather station, 2) humidity (ppmv) measured by the Picarro laser spectrometer; 3) $\delta^{18}$O (‰) in water vapor. Colored lines are ECHAM6-wiso first level outputs (6-hourly resolution): the red (blue) line is representing the mean of oceanic (continental) grid cells.

However the modeled water vapor $\delta^{18}$O signals for oceanic and continental grid cells are very different and neither of them fits the measurements. In particular, the variability of the simulated water vapor $\delta^{18}$O in the oceanic grid cell (std of 2.9 ‰) is lower than in the DDU record (std of 5.2 ‰), while the variability in the continental grid cell is higher (7.1 ‰). Finally, the correlation between data and modeled water vapor $\delta^{18}$O is better for the continental grid cell ($R^2$=0.6) than for the oceanic grid cell ($R^2$=0.4). These results show that the modeled water vapor isotopic composition in this region at the frontier between the Antarctic ice sheet and ocean areas is strongly sensitive to the cell position over the ocean or the continent, more than for variables like temperature or humidity. It follows that both oceanic and continental influences must be considered to interpret our isotopic signal at DDU.

In the following we combine both oceanic and continental grid cells to build a modeled time series of temperature, humidity and water vapor $\delta^{18}$O comparable with our observations. For temperature and humidity, we directly compute weighted averages of the two grid cells time series, while for $\delta^{18}$O we also take into account the humidity content of each air mass to compute the combined weighted isotopic ratio. Three weighting approaches are investigated: a) weights based on the distance between DDU coordinates and grid cell centers, b) weights computed to minimize the distance between measured and modeled mean humidity (Nelder-Mead simplex algorithm as described in Lagarias et al. (1998)) and c) weights computed as in b) but with $\delta^{18}$O as target. If we look at differences between the mean isotopic compositions, standard deviations and the correlation





**Table 2.** ECHAM6-wiso combination (a, b and c, see Text) of first level outputs results and comparison with measurements. Mean and standard deviation of meteorological variables and isotopic composition (left section). Correlations between model and observations (middle section). Weights of each grid cell for the different computation schemes (right section).

| Criteria | Temperature | | Humidity | | $\delta^{18}O$ | | Temp. corr. (ECHAM vs Meas.) | | Hum corr. (ECHAM vs Meas.) | | $\delta^{18}O$ corr. (ECHAM vs Meas.) | | Oceanic (70 m a.s.l.) | Continental (673 m a.s.l.) |
|---|---|---|---|---|---|---|---|---|---|---|---|---|---|---|
| | mean | std | mean | std | mean | std | slope | $R^2$ | slope | $R^2$ | slope | $R^2$ | weight | weight |
| a: distance | -16.2 | 7.9 | 1566 | 1135 | -30.7 | 3.9 | 1.1 | 0.9 | 0.9 | 0.9 | 0.5 | 0.5 | 0.31 | 0.69 |
| b: humidity | -15.1 | 7.9 | 1800 | 1203 | -26.9 | 3.3 | 1.1 | 0.9 | 1.0 | 0.9 | 0.4 | 0.4 | 0.48 | 0.52 |
| c: isotope | -16.7 | 8.0 | 1458 | 1107 | -32.5 | 4.4 | 1.1 | 0.9 | 0.9 | 0.9 | 0.6 | 0.6 | 0.21 | 0.79 |
| Measurements | -11.2 | 7.1 | 1833 | 1196 | -32.3 | 5.2 | | | | | | | | |

coefficients with observations we observe that the three sets of weights improve the comparison between ECHAM6-wiso outputs and measurements at DDU compared to individual grid cells (Tab. 2 and Tab. S1).

In Figure 5 we present the results obtained with the weighting scheme c, targeting the best agreement for $\delta^{18}O$ and leading to the highest correlations coefficients between measurements and model outputs for temperature ($R^2$=0.9), humidity ($R^2$=0.9) and $\delta^{18}O$ ($R^2$=0.6), slopes close to 1 for linear regression between observation and model outputs (1.1, 0.9 and 0.6 for temperature, humidity and $\delta^{18}O$ respectively), and the lowest difference between observed and modeled $\delta^{18}O$ standard deviation (0.8 ‰). Also, modeled d-excess average value (7.8‰) and variability (4.4‰) are close to measurements (Fig. S7). This configura-

tion corresponds to a combination of 79% of continental air masses and 21% of oceanic air masses, which gives less weight to the ocean grid cells than the other two combinations (Tab. 2). This shows that only a strong continental influence can explain the isotopic signal recorded at DDU. At the seasonal scale, the modeled time series issued from configuration c shows similar characteristics (average value, variability) as observed humidity, temperature and water vapor isotopic composition (Tab. 3).

    Then we focus on the two major winter synoptics events of 23 July 2019 and 2 July 2020 identified in the previous section.

Those winter events are associated with peaks in meteorological variables and isotopic values (temperature, humidity and $\delta^{18}O$) close or even higher than summer means in the observations as well as in the model outputs, those peak events of temperature (-5.9°C and -9.9°C), humidity (4130 ppmv and 2930 ppmv) and $\delta^{18}O$ (-22.1 ‰ and -24.8 ‰) (Fig. 5 and Tab. 3). The amplitude of this peaks in ECHAM are also comparable to the ones in measurements. This confirms that ECHAM6-wiso is able to reproduce the variability of the signals even during extreme events.

Finally, we compare ECHAM6-wiso daily means of precipitation isotopic composition with observations. We obtain a weak but still significant correlation coefficient between both data sets ($R^2$=0.3) and the slope of the linear regression is 0.4 ‰ ‰$^{-1}$ (Fig. S8). In particular, the seasonal cycle is well captured both by the observations and model outputs with lower mean $\delta^{18}O$ values during winter and higher mean $\delta^{18}O$ during summer in both modeled and measured precipitation (Fig. S9). The daily precipitation $\delta^{18}O$ samples are however strongly scattered and it is not possible to observe in the precipitation $\delta^{18}O$ record




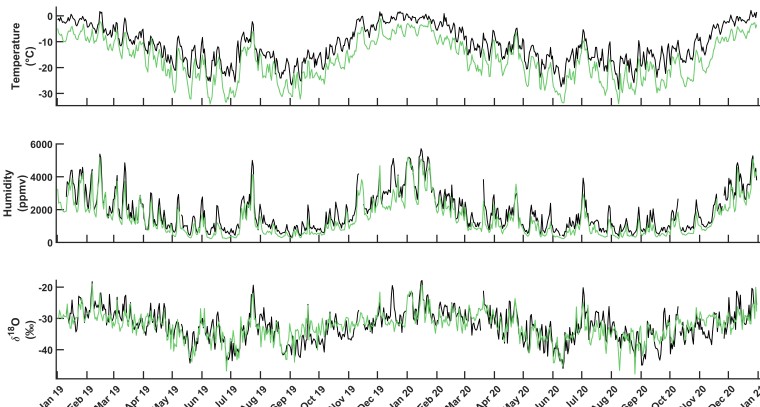

**Figure 5.** Comparison of temperature, humidity and $\delta^{18}$O modeled by ECHAM6-wiso and measured at DDU station over the period 2019-2020. Meteorological and isotopic measurements at DDU at daily resolution are shown in black. Panels from top to bottom: 1) 2m-temperature (°C) from Meteo France weather station, 2) humidity (ppmv) measured by the Picarro laser spectrometer; 3) water vapor $\delta^{18}$O (‰). Green lines correspond to the ECHAM6-wiso combination of surface air level outputs (daily resolution) using configuration c (see text).

**Table 3.** Mean and standard deviation (std) of meteorological variables and isotopic composition calculated from daily means (left). Correlations are calculated with daily data (right). Top: from DDU measurements. Bottom: from ECHAM6-wiso first level outputs using the configuration c) of cell combinations.

| | | Temperature (°C) | | Hum. (ppmv) | | | $\delta^{18}$O (‰) | | d-excess (‰) | | $\delta^{18}$O vs hum. | | $\delta^{18}$O vs temp. | |
|---|---|---|---|---|---|---|---|---|---|---|---|---|---|---|
| | | mean | std | mean | std | relative std (%) | mean | std | mean | std | slope (‰ ppmv$^{-1}$) | R$^2$ | slope (‰ °C$^{-1}$) | R$^2$ |
| **DDU** | DJF | -2.3 | 2.4 | 3373 | 998 | 30 | -27.4 | 3.5 | 10.2 | 3.2 | 2.4 10$^{-3}$ | 0.45 | // | 0.06 |
| | JJA | -16.9 | 5.0 | 1194 | 783 | 66 | -34.7 | 5.0 | 7.8 | 3.3 | 4.5 10$^{-3}$ | 0.51 | 0.64 | 0.42 |
| | Total | -11.2 | 7.1 | 1833 | 1196 | 65 | -32.3 | 5.2 | 8.4 | 3.4 | 3.4 10$^{-3}$ | 0.61 | 0.52 | 0.51 |
| **ECHAM** | DJF | -7.1 | 2.8 | 2815 | 1002 | 38 | -29.1 | 3.2 | 6.5 | 3.1 | 1.9 10$^{-3}$ | 0.4 | // | 0.1 |
| | JJA | -23.1 | 6.2 | 783 | 664 | 88 | -35.7 | 4.4 | 9.3 | 4.8 | 4.2 10$^{-3}$ | 0.4 | 0.47 | 0.4 |
| | Total | -16.7 | 8.0 | 1458 | 1107 | 79 | -32.5 | 4.4 | 7.8 | 4.4 | 2.6 10$^{-3}$ | 0.4 | 0.38 | 0.5 |

(hereafter, $\delta^{18}$O$_p$) an equivalent to the strong peaks observed in the water vapor $\delta^{18}$O during the two strong mid-winter synoptic events (Fig. S9).



The above comparisons between model and observation support the use of ECHAM6-wiso to interpret the isotopic signal variability in water vapor and precipitation at the seasonal and interannual scale in the surrounding of the DDU station. It is thus a very powerful tool to help interpreting snow and ice cores in the region.

## 4   Discussion

The water isotopic series shown above have highlighted the complex relationships between temperature and $\delta^{18}O$ of vapor and precipitation at DDU. We have also shown that the atmospheric model equipped with water isotope ECHAM6-wiso reproduces daily and seasonal mean and variability of the isotopic observation records. As a consequence, we explore here how this model can help interpreting firn cores isotopic records in Adelie Land. Here, we focus on the S1C1 firn core analyzed by Goursaud et al. (2017). This 22.4 m core has been drilled during the 2006/2007 season at 66.71°S, 139.8°E, 279 m a.s.l., 15 km from DDU (Fig. 1). It covers 60 years with a mean accumulation rate of $218 \pm 69 \ \mathrm{kg \, m^{-2} \, yr^{-1}}$ but we will limit our study to the period back to 1979 in order to be able to compare with reanalyses data. The S1C1 $\delta^{18}O$ record measured with a resolution of 5 cm (Fig. 6a) shows variations with a maximum amplitude of $\delta^{18}O$ variations of 10 ‰. There is no clear annual periodicity in the $\delta^{18}O$ as seen in the frequency spectrum (Fig. 6b).

We create virtual firn cores (VFC) following the approach of Sime et al. (2011) to study the origin of the $\delta^{18}O$ variations in the S1C1 core (Text S1). The first VFC is obtained using temperature and precipitation from ERA5 (red curves in Fig. 6): $\delta^{18}O$ is directly linked to temperature weighted by precipitation to account for precipitation intermittency. For this construction, we used the slope of $0.44 \ ‰ \ °C^{-1}$ observed at DDU between $\delta^{18}O$ of precipitation and temperature (Fig. S10). This first VFC, hereafter called VFC-ERA5, displays clear annual cycles of $\delta^{18}O$ with an average amplitude of 7‰ (Fig. 6a). The standard deviation of the signal remains stable over time which is different from the S1C1 $\delta^{18}O$ record showing large variations of the standard deviation with depth (Fig. S11a). The frequency spectrum indeed reveals a marked annual periodicity (Fig. 6b).

The poor agreement between VFC-ERA5 and S1C1 $\delta^{18}O$ records (Fig. 6a and 6b) clearly confirms that temperature is not the only driver of $\delta^{18}O$ in this Adelie Land firn core. We also produced a second VFC based on the outputs of ECHAM6-wiso (green curves in Fig. 6). In this case, the $\delta^{18}O$ record of VFC-ECHAM is calculated from the $\delta^{18}O$ of precipitation and precipitation amount, both simulated by ECHAM6-wiso back to 1979 and provided at a 6-hourly resolution. As for the VFC-ERA5, the VFC-ECHAM record displays a clear annual cycle visible in the associated spectrum (Fig. 6b). However, it also displays a much larger variability in the amplitude of the annual cycle than the VFC-ERA5. This results in larger variations of the moving standard deviation of VFC-ECHAM $\delta^{18}O$ record which looks like variations observed in the temporal evolution of the standard deviation of S1C1 $\delta^{18}O$ record (Fig. S11a). There is thus, in general, a better agreement of S1C1 $\delta^{18}O$ record with VFC-ECHAM $\delta^{18}O$ record than with VFC-ERA5 $\delta^{18}O$ record. Still, the amplitude variations of the standard deviation of VFC-ECHAM $\delta^{18}O$ record don't show a clear decrease tendancy in comparison to what is observed for S1C1 $\delta^{18}O$ record, a difference which may be attributed to the diffusion of water isotopes within the firn.

In Figure 6c, the influence of isotopic diffusion in the firn has been considered for the construction of the VFC records (details in Text S2). The rapid decrease of the amplitude of the annual cycles for increasing depth in S1C1 is reproduced by




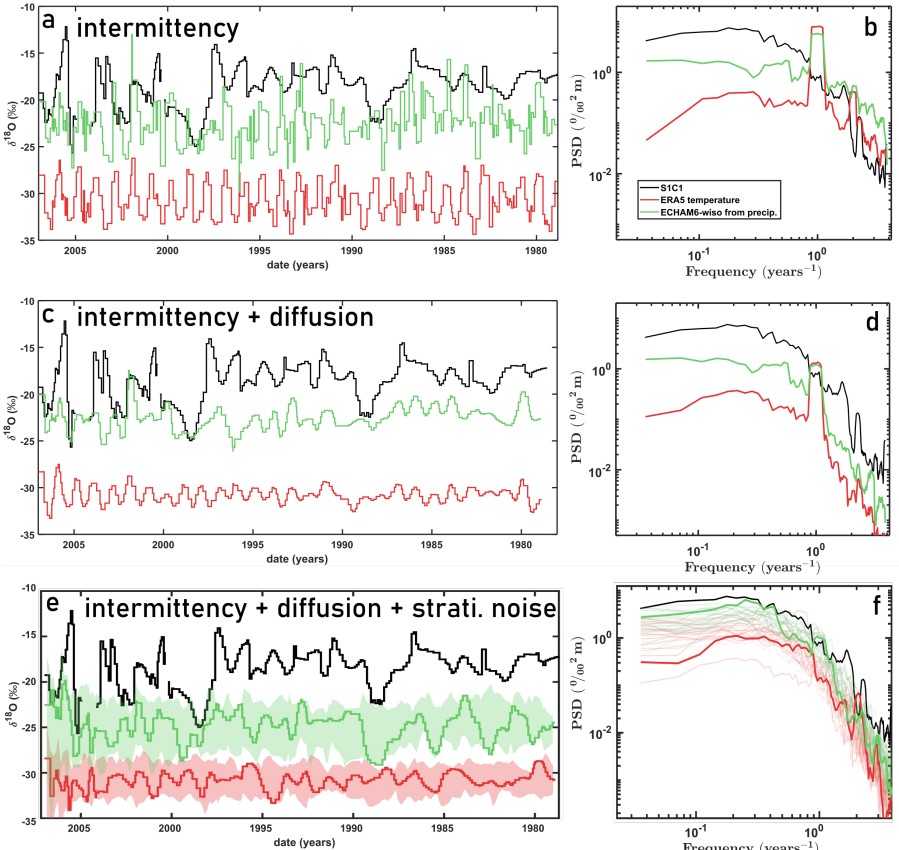

**Figure 6.** In black, isotopic composition ($\delta^{18}$O in ‰) of S1C1 (first 620 cm of water equivalent) firn core versus date (years) using the dating based on chemical analyses in Goursaud et al. (2017). Red and green curves are the isotopic compositions of VFC resampled at S1C1 sampling resolution and based on temperature from ERA5 and ECHAM6-wiso isotopic composition of precipitation respectively (see text). In the first row, a) and b) VFCs are calculated with precipitation intermittency. In the second row, c) and d) VFCs are built with precipitation intermittency and isotopic diffusion. In the third row, e) and f) VFCs are calculated with precipitation intermittency, isotopic diffusion and simulated stratigraphic noise (25 random noise simulations). An artificial bias is added to VFCs (-5‰ and -10‰, respectively for green and red curves) in order to improve readability. Right column presents the associated frequency spectrums. On the right column, b), d) and f) spectrums are computed using Thomson's multitaper power spectral density. Note that for the last row (precipitation intermittency + isotopic diffusion + simulated stratigraphic noise), one stratigraphic noise simulation result is highlighted in bold (over the 25 simulations), other simulations are represented through shaded envelops (e) or light plots (f).

this VFC. This effect is also seen in the associated spectrum (Fig. 6d) with a decrease of the power in the high frequency range (from 1.2 to 0.04 ‰ m$^2$ in average for frequencies greater than 1.1 yr$^{-1}$, in ECHAM VFC), below the level of S1C1 spectrum. Diffusion has also the effect of strongly decreasing the amplitude of the standard deviations of the VFC-ERA5 and VFC-ECHAM records (Fig. S11, Tab. S2), again below the level observed in the S1C1 record. This may be due to a too





strong diffusion effect in our approach. Improvement of the diffusion model may be able to reconcile the amplitude of the high
frequency variability of the VFC $\delta^{18}$O signals with the amplitude of the high frequency variability of the S1C1 $\delta^{18}$O signals. It
may however not explain why there is no clear temporal correspondence between S1C1 $\delta^{18}$O signal and the VFC $\delta^{18}$O signals,
and why the annual cycle is not seen in the S1C1 signal, which may be explained by errors in the chronology of the S1C1 cores.
Another point of disagreement is the lower power associated with low frequency signal in the VFC records than in the S1C1
signal. These three points of disagreement may be caused by deposition or post-deposition effects creating a non climatic low
frequency variability while destroying the record of the annual variability (through wind blowing for example). To simulate
such an effect, we follow Laepple et al. (2018) by adding white stratigraphic noise that is controlled by two parameters: (i) the
relative amount of noise compared to the input signal (0 % to 100 %) and (ii) the length at which the noise impacts the signal
(from 1 to 10 cm). In Figure 6e, we show that using a noise level of 90 % and a noise scale of 7 cm, the VFC-ECHAM inter-
annual amplitude variability is more likely to match S1C1 signal. This is confirmed by spectral analyses (Fig. 6f), where the
difference between S1C1 and ECHAM VFC spectrums become comparable (4.1 and 2.3 ‰ m$^2$ in average for low frequencies
i.e. from 0.1 to 0.5 years$^{-1}$, respectively for S1C1 and ECHAM, for example). We also notice that even with a noise level of
100 % (maximum authorized in this model), the ERA5-VFC spectrums hardly improve in comparison with observation (0.9 ‰
m$^2$ in average for low frequencies, for example). Note that invoking a strong stratigraphic noise level to explain the difference
in the spectrum obtained for S1C1 and the VFC cores is not the only explanation. The linear relationship estimated between
$\delta^{18}$O of vapor and precipitation at DDU shows that precipitation only imprints part of the vapor signal variability (slope =
0.6 ‰‰$^{-1}$ with R$^2$=0.4 in ECHAM6-wiso, see Fig. S12). A too low interannual climatic variability in the $\delta^{18}$O of model
precipitation would also explain the lower power in the low frequency range of the spectrums of the VFC records.

As a conclusion, we have shown that the use of the ECHAM6-wiso allows to better explain the firn core $\delta^{18}$O variability than
with the only influence of temperature. Still, the interannual $\delta^{18}$O variability simulated by ECHAM6-wiso is not able to explain
the low frequency variability of the firn core $\delta^{18}$O in Adelie Land. This may be due to an underestimation of the interannual
climatic or $\delta^{18}$O variability in the model and/or to stratigraphic noise associated with deposition and post-deposition effects.
In order to cancel the influence of the stratigraphic noise, we would need to stack several firn core (Münch and Laepple, 2018),
in order to increase the signal to noise ratio (Laepple et al., 2016, 2018; Casado et al., 2020).

## 5 Conclusions

We present the first multi-year continuous record of isotopic composition in surface vapor and precipitation at Dumont d'Urville
station (Adélie Land), a coastal site in East Antarctica. This region is characterized by the presence of strong katabatic wind
and the local influence of ocean and sea ice. This new dataset allows us to characterize diurnal variations of meteorological
variables (temperature, humidity and $\delta^{18}$O) for the different seasons and to determine the evolution of key relationships ($\delta^{18}$O
versus temperature or humidity) along the year. We found mean annual slopes of 0.5 and 0.4 ‰ °C$^{-1}$ for daily $\delta^{18}$O vs.
temperature in the water vapor and in the precipitation respectively. The warm and wet synoptic events occurring in winter
and associated with strong precipitation are clearly imprinted in the water vapor isotopic signal while our precipitation water





isotopic signal only captures the strong seasonal cycle. We evaluate the ECHAM6-wiso model through a comparison of the simulated $\delta^{18}$O of water vapor and precipitation with our record and we show that a combination of continental (79 %) and oceanic (21 %) grid cells leads the modeled temperature, humidity and $\delta^{18}$O to nicely fit the observations. Winter extreme

synoptic events are also correctly represented by the model. The excellent agreement between modeled and measured isotopic series encourages us to investigate how ECHAM6-wiso could help understanding isotopic signals recorded in ice-cores in the region. We focus on the S1C1 firn core previously studied and located 10 km inland from DDU. Constructing virtual firn cores from modeled temperature and water isotope and taking into account the precipitation intermittency, we show that a pure temperature interpretation fails in explaining the interannual variability observed in the measured S1C1 isotopic record. We

improve the agreement between measured and modeled records when using the water isotopic composition of precipitation instead of temperature, but the low frequency variability is still underestimated. Are results suggest that either the $\delta^{18}$O of precipitation in ECHAM6-wiso model is not expressing enough variability at interannual to decadal scales or that deposition and post-deposition effects contribute significantly to the isotopic signal recorded in the S1C1 core drilled in Adelie Land. Disentangling these two effects may be done by stacking several firn core records in the region to get rid of the stratigraphical

noise.

*Data availability.* The full dataset (isotopic composition of vapor and precipitation and meteorological data) archiving is underway and will be fully available through a Zenodo repository: 10.5281/zenodo.7708489. Until publication, the data set is with limited access.

*Acknowledgements.* This work was supported by the IPEV ADELISE project, the LEFE program ADELISE. This article is part of the ANR ARCA (Atmospheric River Climatology in Antarctica; grant number ANR-20-CE01-0013) funded by the French National Research Agency.

It was also funded by the Fondation Prince Albert 2 de Monaco under the project Antarctic-Snow. We thank Valérie Masson-Delmotte that contribute to this study by providing a Picarro analyser. We acknowledge the staff from IPEV (especially Quentin Perret), all the winter-overs that contribute to setup maintenance (especially Audrey Teisseire, Gregoire Aufresne, Guillaume Herment) and the Meteo France members for their avaibility (especially Gaetan Heymes).

*Supplement.* The supplement related to this article is available on-line at: DOI/XXX



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
