# Peer review of "From atmospheric water isotopes measurement to firn core interpretation in Adelie Land: A case study for isotope-enabled atmospheric models in Antarctica"

_EGUsphere, 2023_

## Referee Comment (RC1)

**Review Leroy-Dos Santos et al "From atmospheric water isotopes measurement to firn core interpretation in Adelie Land: A case study for isotope-enabled atmospheric models in Antarctica"**

The manuscript entitled "From atmospheric water isotopes measurement to firn core interpretation in Adelie Land: A case study for isotope-enabled atmospheric models in Antarctica" by Leroy-Dos Santos et al. presents the first 2-year continuous isotope record (d18O, dD) of atmospheric water vapor and daily precipitation at a coastal site in East Antarctica. The authors show that observed seasonal and interannual isotope variability of atmospheric water vapor and precipitation are well captured by an isotope-enabled atmospheric general circulation model (ECHAM6-wiso), considering the contribution of oceanic and continental influences. They demonstrate that the ECHAM6-wiso allows to better explain the isotope signal of ice cores at coastal sites, considering not only temperature, but also precipitation intermittency, diffusion and post-depositional effects. The results of this study have important implications for the interpretation of firn and ice core data in terms of paleotemperatures.

The methods and results are well presented, and the discussion is detailed, well-written and reasonable. I have two general comments, where I would like the authors to elaborate a bit more in detail:

1) The authors show higher frequency of synoptic events in winter. Can the authors specify what the mean with "synoptic events". They state that synoptic events are seen as meridional exchanges, bringing warm moist air masses to the site, but are they always associated with precipitation and are they the only process causing on-site precipitation? What are the reasons leading to these synoptic events? Are they really less frequent in summer or just harder to identify in the meteorological record? It would be great if the authors could elaborate a bit more on these synoptic events, as it seems to be a major driver of d18O of vapor and precipitation and thus also relevant for the interpretation of ice core data.

2) The authors show seasonal variations in the relationship between d18OV, T and humidity. Can the authors further evaluate, which processes are responsible for these changes and its relevance for the interpretation of ice core data?

Further, there are a number of passages that may be reformulated to be more precise and concise. Some of them are listed below. I would encourage the authors to take care of using concise language when revising the manuscript.

**Minor comments / recommendations:**

**Abstract**

Line 7: Can you specify the isotope composition of which compartment of the water cycle do you mean? Do you want to refer here to ice core records?

Line 9: You should specify that "humidity" refers here to the atmospheric water mixing ratio not to atmospheric relative humidity.

Line 11: You may want to state here if you also found a relationship to humidity. And, what does these relationships between d18O, T, and humidity imply? Is there a relation between the

d18O vs T relationship and the d18O vs humidity relationship (i.e. do T and humidity correlate)?

Line 17: Instead of giving an outlook can you specify how this link between vapor and precipitation helps to interpret short firn cores?

**Introduction**

Line 34: Remove "providing".

Line 51: Remove "record".

Line 53: I think there is no need to start a new paragraph here.

Line 69: In the named study, comparison of water vapor isotope data and back trajectory simulations was performed to … ?

Line 71: suggested change to "the instrumental setup and the 2-year isotopic series". Remove "analysis of a".

Line 72: Suggested change to "Then, we show the ECHAM6-wiso outputs […]"

Line 73: Remove "evaluated".

**Methods**

Line 85: The start of this sentence sound strange to me. I suggest to avoid "in the following study we discuss" and reformulating to be more concise, e.g. "The laser spectrometer measures molecular water vapor mixing ratio and …"

Line 92: To be more concise, I suggest: "The d18O and dD series were calibrated following …"

Line 102: Please specify "isotope-humidity relationship". You may also highlight that you do not observe a difference in this relationship in the field and at LSCE.

Line 116: Can you specify how many replicates per sample? (2-3?). You do not use an independent standard that you measure routinely in each sequence that you could use to estimate uncertainty?

Line 123-125: Can you specify for what these data are used? (Comparison to isotope data? / Identification of processes driving isotope variability of atm water vapor/precipitation / input for ECHAM6-wiso model). If you use the ERA5 reanalysis data solely for nudging the ECHAM6-wiso model, you may consider combining this section with the following.

**Results**

Table 1: Specify that the isotope composition of atmospheric water vapor is shown, not for precipitation. Verify throughout the paper if there is a need to specify if you mean atm water vapor or precipitation.

Table 1: Why there is no value for the slope between d18O vs Temp in DJF? I think something shifted in this table (cf. Fig. S4). Please check.

Line 147: According to Table 1, also d-excess shows higher values in summer (10.2) than in winter (7.8), but the difference is not significant?

Line 153: "[…] lead to higher variability […] when there are larger meridional temperature and moisture gradients than in summer."

Line 167: I can't find these numbers in Table 1. Please check if something shifted. You may also consider showing the plots for the full period in Fig S3 and S4. Also, I don't think that a slope of 0.5 ‰°C$^{-1}$ is significantly different from 0.6 ‰°C$^{-1}$. In contrast, the slope seems to be significantly higher in fall and winter than in spring and summer. Please verify.

Line 169: Is it the synoptic events that are not visible or rather the impact of the synoptic events on the meteo data?

Line 170: The observation of diurnal cycle only in summer is very interesting. However, std given in Table 1 is based on daily means. Hence, diurnal variability should not affect this value, shouldn't it?

Line 202: Remove "temperature at DDU".

Line 237: You may consider highlighting the two precipitation peak events in figure 5.

Line 238: "The amplitude of these peaks in ECHAM is comparable to that of measurements."

Table 3: Consider combining with Table 1, as half of the presented information are identical.

**Discussion**

Line 253: Suggested change to: "In the following, we explore how […]"

Line 254: Suggested change to: "For this, we use […]"

**Conclusion**

Line 319: I do not see the nice agreement (in absolute values) for all three parameters simultaneously. Approach c fits d18OV, but produces an offset of 400 ppmv in humidity, while temperature is always about 4-5°C underestimated by the model. Irrespective of the offset in absolute values, however, the observed trends are well reproduced by the model. You may want to specify this here. Also, you do not show model results for approach a and b, so that the reader remains unknown if these model approaches also allow to reproduce general trends. You may do so in the supplement.

Line 326: Typo. "Our results", not "are results".

---

## Author Comment (AC2)

**From atmospheric water isotopes measurement to firn core interpretation in Adelie Land: A case study for isotope-enabled atmospheric models in Antarctica**

**ANSWER TO REVIEWERS:**

We thank the reviewers for their very interesting comments. They motivated the clarification of synoptic events definition and the reinforcement of their potential impact on isotopic signals during winter. It is key for firn core interpretation and we rewrote and reordered the 3.1 section in order to improve the quality of this analysis. The isotopic data series correction will be clarified and the abstract will be re-written as explained below. The other comments will also be taken into account in the new version of the manuscript as explained below.

Reviewer #1:

1) **The authors show higher frequency of synoptic events in winter. Can the authors specify what the mean with "synoptic events". They state that synoptic events are seen as meridional exchanges, bringing warm moist air masses to the site, but are they always associated with precipitation and are they the only process causing on-site precipitation? What are the reasons leading to these synoptic events? Are they really less frequent in summer or just harder to identify in the meteorological record? It would be great if the authors could elaborate a bit more on these synoptic events, as it seems to be a major driver of d18O of vapor and precipitation and thus also relevant for the interpretation of ice core data.**

We agree that synoptic event definition and detection was unclear in the manuscript. Our study focuses on the main precipitation events, as they are ultimately responsible for the isotopic signal recorded in the firn cores. Firstly, we identify these main precipitation events and conclude that they must be linked to the intrusion of warm, moist air masses from the north, as they are often associated with increased humidity and temperature on a daily time scale, corresponding to synoptic-scale events. Secondly, we observe that the occurrence of these events does not depend on the season (comparison DJF vs JJA), but that their impact on local weather conditions or climatology does. The anomalies (temperature, d18Ov) caused by these intrusions are larger in winter and could have a significant impact on winter variability, as identified by (Servettaz et al., 2020) at ABN.

We will re-write the corresponding paragraph in 3.1 section with the arguments detailed in the previous paragraph in order to address those comments.

2) **The authors show seasonal variations in the relationship between d18OV, T and humidity. Can the authors further evaluate, which processes are responsible for these changes and its relevance for the interpretation of ice core data?**

At DDU, the impact of synoptic events in different weather dynamics (summer diurnal cycles versus winter synoptic variability) are sufficient to explain seasonality of relationships.

In winter the temperature range is larger than in summer: this can be seen in std in Table 1 (std for temperature is doubled in winter compared to summer) and in Figure S4. This is due to the different impact of the synoptic events regarding the season: the rapid switch of air origin in winter will drive the relationship between d18OV and temperature. In summer, the influence of the intrusion of a warm air mass is not very visible because the background climate is already warm.

As for the humidity to d18OV relationship, the higher slope found in winter compared to summer one is expected (see Table 1 and Figure S3) from the relationships between d18O and humidity (or mixing ratio) along distillation line or during mixing of 2 different air masses. We provide below a figure where we plot the relationship between d18O and humidity for 2 processes: 1) pure Rayleigh distillation and 2) mixing of two air masses

[Figure]

Figure 1: Relationship between the 18O and humidity for the vapor phase in a Rayleigh distillation (red), an air mass mixture (green)

It follows from this figure that the d18O vs humidity slope is larger at low humidity (winter) than at high humidity (summer) as observed in the data. We will give this explanation in the new version of the manuscript.

In the new version of the manuscript, we will reorder part of the 3.1 section in order to reinforce the link between the different weather regimes in summer and winter and the relationships observed between d18Ov variations and climatic parameters. We will discuss both winter and summer weather regimes before we conclude on the different slopes. Then we will argue on the rapid air origin switch to argue on the seasonal differences of the d18O to temperature relationships. In addition, we will add a comment in the manuscript on the impact of slope seasonality on the interpretation of temperature proxies in ice cores: we need to investigate the link between isotopic composition of vapor and precipitation to study the impact on ice-core interpretation as we classically use a unique slope (site dependent) to convert isotopic signal into temperature.

Reviewer #2:

1) **The abstract is a bit cursory without concision, and lacks some important conclusions. I would suggest the authors rewrite it.**

   We agree with the reviewer #2 that the abstract could be significantly improved. We will rewrite it and add the following conclusions: 1) ECHAM6 wiso is able to reproduce isotopic signal at DDU at different time scales in water vapour (synoptic and seasonal scales) 2) VFCs built from ECHAM6-wiso data show lower frequency variability than observed (S1C1 firn core): deposition and post deposition effects contribute significantly to the isotopic signal recorded in coastal sites, as well as probably an ECHAM underestimated variability at interannual to decadal scales of precipitation amount and/or isotopic composition.

2) **The description of the isotopic calibration is lacking detail and uncertainties. The plots of drift correction should be provided in SI. Delta 18O-humidity calibration data from SDM are missed in Fig. S2.**

   Indeed, there's a problem on figure S2, the data made with SDM are plotted but they are not plotted in the right color (dark gray circles), which explains the confusion, Figures have been replotted and replaced in the Supplementary material.

   Regarding the drift correction, in the next manuscript version (in supplementary material), we will add the Figure displayed below showing drift estimation through 48h period routine automatic measurements in comparison with the drift estimated through 4 different humidity calibration sessions. Also, we will comment those results in a paragraph (in the supplementary online material) in order to explain why we do not consider drift correction in regard to the humidity correction applied to the data series. A first proposition for this explanation is given below.

[Figure]

[Figure]

Text SXX: Mean linear drift estimation from different sets of data. Black crosses are routine measurements and red crosses are measurements performed during humidity calibration sessions. Grey bars are the standard deviation associated with each measurement. Black (red) line is the linear drift estimated from routine measurements (humidity calibration sessions measurement). The green line is the linear drift estimated from both data series.

Standard measurements (black crosses) was performed every 2 days with the humidity generator (humidity was set at 1140 ppmv on average, 40 minutes' duration for each measurement level). Some technical issues led us to select only 150 calibrations over the 2 years' period. The results show a drift with decreasing d18O and dD values with time. Unfortunately, the data are very scattered and even sparse after the first year of installation of the instrument. The reason for this scattering is a problem with the humidity generator (bad drying procedure in the instrument) when it was working without human intervention. However, we could perform proper calibrations each year during the field summer seasons (red crosses). Because we are more confident with these measurements, we have only kept these series for the drift estimation.

Mean drift over the two-year period is estimated to 0.01 ‰/years and 0.6 ‰/years for $\delta^{18}O$ and $\delta D$, respectively. Because the drift is very small but associated with a high uncertainty, we decided not to correct our data series from mean annual drifts but to associate a large uncertainty with d18O and dD. The uncertainty associated with δ18O and δD measurements in water vapor is calculated as the 70th percentile of the distribution of the 4 annual calibration during the summer season and results in 0.8 ‰ and 3.2‰, respectively. Note that the new version of the LHLG (low-humidity level generator), installed in January 2022 at DDU, does not show any more the scattered patters in the routine calibrations performed on a 48h periodicity and that there is a good agreement between the drift inferred from this routine calibration and the drift calculated from the calibrations performed during the summer season. We will provide these explanations as well as the figures in the supplementary material of the new manuscript.

**3) The influence of synoptic events does not address very well. Some descriptions are not clear at seasonal or events scales. For instance, what is the difference of such events between winter and summer, and what is the reason for those distinct influences in L150-155? How could get the conclusion in L315-317?**

Thanks for this comment which echoes that of reviewer #1. Therefore, we copy below the answer to the similar comment.

It seems that our definition and detection of synoptic events is, indeed, unclear and this makes our result and interpretation confusing. Here we are interested in main precipitation events, as in definitive, they are responsible for isotopic signals recorded in firn cores. First, we identified those main precipitation events (using a daily precipitation rate as described in the manuscript) and then we conclude that they must be related to warm and moist air masses intrusion coming from North (as they are really often associated to humidity and temperature increase on a daily time scale), corresponding to synoptic scale events. Then, we observe that occurrences of such events aren't seasonally dependent (DJF vs JJA comparison) but their impact on local weather or climatology is. Indeed, as pointed by reviewer #1, the anomalies (temperature, d18) caused by those intrusions are more important in winter and could significantly impact winter variability, as identified by (Servettaz et al., 2020) at ABN.

We will re-order and partially re-write the 3.1 section in order to address those comments and the comment #1 from reviewer #1.

Also, this impact asymmetry of synoptic events in regard to season is reflected in relationship between d18 and temperature for example. This will be addressed in comment #2 of reviewer #1. In the modified manuscript, we comment on the rapid air origin switch to argue on the seasonal differences of the d18O to temperature relationships.

Regarding the last part of this comment, we do believe that our new analyses, thanks to the reviewers comments are clearer and that they are not contradictory with what is written in the conclusion: "The warm and wet synoptic events occurring in winter and associated with strong precipitation are clearly imprinted in the water vapor isotopic signal while our precipitation water isotopic signal only captures the strong seasonal cycle.". While winter synoptic events are clearly observable in water vapour signal, the seasonal signal is observable in both vapour and precipitation isotopic signal (see mean values in Table 1).

**4) The evaluation of d-excess from ECHAM6-wiso is missing in section 3.2. The d-excess variability is well established from observations, but does not show any related analysis to combine with simulations. Why?**

We thank the reviewer for this comment. Indeed, there is a lack of comment on the comparison between measured d-excess and d-excess issued from the ECHAM6-wiso model even though comparison is made on Table 3 and Figure S7. From those

data, we observe that ECHAM6-wiso is not able to reproduce the second-order parameter d-excess. In fact, apart from the comparable mean values over the two-year data series (8.4 ‰ and 7.8 ‰ respectively for measurement and model output), the model fails in reproducing the seasonality of observed d-excess. Seasonality is actually inverted in the model: while d-excess reaches its maximum in summer (JJA) in observations (10.2‰), its value is minimal in ECHAM6-wiso combination (6.5‰).

We will add this short analysis to the revised manuscript in section 3.2.

**5) I can not get the point clearly in L243-246. It seems controversial with the conclusion.**

"In particular, the seasonal cycle is well captured both by the observations and model outputs with lower mean δ18O values during winter and higher mean δ18O during summer in both modeled and measured precipitation (Fig. S9). The daily precipitation δ18O samples are however strongly scattered and it is not possible to observe in the precipitation δ18O record (hereafter, δ18Op) an equivalent to the strong peaks observed in the water vapor δ18O during the two strong mid-winter synoptic events (Fig. S9)."

We understand that this paragraph was not clear enough. We thus propose to add the following explanations at its end.

"Because the sampling of precipitation was limited to one sample per day and only for the days with precipitation, it is expected that we cannot observe the same d18O signal in the precipitation record than in the continuous water vapor at an hourly resolution."

Moreover, we do believe that this paragraph is not contradictory with what is written in the conclusion: "The warm and wet synoptic events occurring in winter and associated with strong precipitation are clearly imprinted in the water vapor isotopic signal while our precipitation water isotopic signal only captures the strong seasonal cycle."

**1) Increase all font sizes in figures and figures are too small to see details.**

We have increased the font size in all figures in the manuscript.

**2) It is hard to compare the vapor data and precipitation data when they are plotted on separate panels with distinct axis scales in Fig. S9.**

Indeed, the figure S9 does not permit one to easily make this comparison. While comparing signal imprinted in vapor in comparison to precipitation, we should refer to Figure S12. This figure is a scatter plot comparing isotopic signal in vapor versus isotopic signal in precipitation for both model and observation. We will change the referencing in the modified manuscript.

---

## Author Response (AR1)

**From atmospheric water isotopes measurement to firn core interpretation in Adelie Land: A case study for isotope-enabled atmospheric models in Antarctica**

**ANSWER TO REVIEWERS:**

We thank the reviewers for their very interesting comments. They motivated the clarification of synoptic events definition and the reinforcement of the potential impact on isotopic signals of those during winter. It is key for firn core interpretation and we rewrote and reordered the 3.1 section in order to improve the quality of this analysis. The isotopic data series correction will be clarified and the abstract will be re-written. As for other comments, we have resolved all other comments in the following.

You will find my answer to comments in blue and the modification induced in the manuscript in red in this documents. You will also find a document (named 2023_clds_et_al_ddu_wiso_19_20-diff_bis.pdf) where the differences between the manuscript versions are highlighted (the old text is in red and crossed out, while the new text is in blue).

**Reviewer #1:**

1) **The authors show higher frequency of synoptic events in winter. Can the authors specify what the mean with "synoptic events"? They state that synoptic events are seen as meridional exchanges, bringing warm moist air masses to the site, but are they always associated with precipitation and are they the only process causing on-site precipitation? What are the reasons leading to these synoptic events? Are they really less frequent in summer or just harder to identify in the meteorological record? It would be great if the authors could elaborate a bit more on these synoptic events, as it seems to be a major driver of d18O of vapor and precipitation and thus also relevant for the interpretation of ice core data.**

We agree that synoptic event definition and detection was unclear in the manuscript. Our study focuses on the main precipitation events, as they are ultimately responsible for the isotopic signal recorded in the firn cores. Firstly, we identify these main precipitation events and conclude that they must be linked to the intrusion of warm, moist air masses from the north, as they are often associated with increased humidity and temperature on a daily time scale, corresponding to synoptic-scale events. Secondly, we observe that the occurrence of these events does not depend on the season (comparison DJF vs JJA), but that their impact on local weather conditions or climatology does. The anomalies (temperature, humidity, d18Ov) caused by these intrusions are larger in winter and could have a significant impact on winter variability, as identified by (Servettaz et al., 2020) at ABN.

In the manuscript changes have been made in the section 3.1. Re-writing of the paragraph introducing events associated with synoptic scale dynamics:

The variability of temperature, δ18O and humidity also shows a seasonality (Tab. 1), with higher standard deviation in
winter than in summer. These results can be explained by a different seasonal impact of the main precipitation events. Blue bars in Figure 2 show the distribution of the 3-day periods centered on daily precipitation rates higher than 4.5 kgm−2 day−1. We detect 35 precipitation peaks over the 2-year period, 7 (20 %) during DJF (summer) and 8 (23 %) during JJA (winter), so no seasonality of the occurrence is observed. Though, the temperature anomalies associated with those events are more important in winter (Fig. S4). Servettaz et al. (2020) demonstrated the key role that such precipitation events, controlled by synoptic scale dynamics, could play in ice-core interpretation at high accumulation sites. As they are associated with warm and moist air intrusions, they cause warm anomalies compared to the seasonal mean. Here, we point out two major events that are particularly intense during extended winter (from May to September): (a) 23 July 2019, with a precipitation rate of 31 kgm−2 day−1 and (b) 2 July 2020, with a precipitation rate of 21 kgm−2 day−1. These events correspond to the largest daily precipitation rates of each winter, and to the first and third largest daily precipitation rates when considering the whole 2019-2020 period. The values of temperature, humidity and δ18O during these winter events (-1 ∘C and -4.4 ∘C, 5780 ppmv and 4370 ppmv, -17.8‰ and -19.3 ‰, respectively for 165 the two events) are close or above summer averages (Tab. 1).

2) **The authors show seasonal variations in the relationship between d18OV, T and humidity. Can the authors further evaluate, which processes are responsible for these changes and its relevance for the interpretation of ice core data?**

At DDU, the different weather dynamics (summer diurnal cycles versus winter synoptic variability) are sufficient to explain seasonality of relationships.

In winter the temperature range is larger than in summer: this can be seen in std in Table 1 (std for temperature is doubled in winter compared to summer) and in Figure S4. This is due to the different impact of the synoptic events regarding the season: the rapid switch of air origin in winter will drive the relationship between d18OV and temperature. In summer, this is almost annihilated.

As for the humidity to d18OV relationship, the higher slope found in winter compared to summer one is expected (see Table 1 and Figure S3). See this figure where we plot the relationship between d18O and humidity for 3 processes: 1) pure Rayleigh distillation, 2) mixing of two air masses and 3) MCIM (Mixed cloud isotopic model)

[Figure]

Figure 1: Relationship between the 18O and humidity for the vapor phase in a Rayleigh distillation (red), an air mass mixture (green) or in the MCIM model. Initial conditions for Rayleigh model: [T = 20°C, 18O = -10.8 ‰, D = -74.9 ‰] . Parametres for MCIM: Winkler et al. (2012)

We found in this figure that there are roughly two regimes, one for low humidity level and one for high humidity.

However, the previous comment (#1) will help highlight the different impact of the synoptic events (associated with main precipitation events) with regard to season and thus partly address this question. In addition, we will reorder part of the 3.1 section in order to reinforce the link between the different weather regimes in summer and winter and the main relationships. We will discuss both winter and summer weather regimes before we conclude on the different slopes. Then we will argue on the rapid air origin switch to argue on the difference before d18O to temperature relationships. We will not argue for humidity as it is classic. In addition, we will comment in the manuscript on the impact of slope seasonality on the interpretation of pure temperature from ice cores: we need to investigate the link between isotopic composition of vapor and precipitation to study the impact on ice-core interpretation as we classically use a unique slope (site dependent) to convert isotopic signal into temperature.

In the manuscript changes have been made in the section 3.1. Re-writing of the paragraph commenting those relationship slopes:

Differences between winter and summer weather regimes impact the relationship between variables in vapor. First, humidity and δ18O show high correlation coefficients (calculated from daily means) both over the whole record (R2=0.6) and at a seasonal scale (R2=0.5 for DJF and JJA). The slope of this linear relationship (Tab. 1 and Fig. S5) is almost doubled during winter (4.5.10−3 ‰ ppmv−1) compared to summer (2.4.10−3 ‰ ppmv−1). This difference between low and high humidity regimes (during winter and summer, respectively) is expected from the relationships between δ18O and humidity along distillation line or during mixing of 2 different air masses (Steen-Larsen et al., 2017). Second, the linear relationship between δ18O and temperature is strong for the full period (R2=0.5) but vanishes during summer (R2<0.1). This can be related to the smaller daily variability during summer, in

comparison to winter when synoptic event occurrences lead to larger increase of temperature and δ18O over synoptic time scale. The δ18O-temperature slope (Tab. 1 and Fig. S6) over the full period (0.5 ‰ ◦C−1) is similar to the winter mean slope (0.6 ‰ ◦C−1, R2=0.4). We note that spring mean slope is slightly different (0.4 ‰ ◦C−1, R2=0.3) but is statistically less representative in comparison. Further, we need to investigate the link between the isotopic composition of vapor and precipitation to study the impact on ice-core interpretation. The condensation of vapor in the upper atmospheric layers leads to precipitation but subsequent exchanges between atmospheric water vapor and snow flakes can also affect the isotopic composition of the collected precipitation. Classically a unique slope (site dependent) is used to convert isotopic signal into temperature.

Minor comments / recommendations:

Abstract

Line 7: Can you specify the isotope composition of which compartment of the water cycle do you mean? Do you want to refer here to ice core records?

We need to progress in our understanding of the influence of the atmospheric hydrological cycle on the water isotopic composition of ice-core.

Line 9: You should specify that "humidity" refers here to the atmospheric water mixing ratio not to atmospheric relative humidity.

We characterize diurnal variations of meteorological parameters (temperature, atmospheric water mixing ratio (hereafter humidity) and δ18O) for the different seasons and [...]

Line 11: You may want to state here if you also found a relationship to humidity. And, what does these relationships between d18O, T, and humidity imply? Is there a relation between the d18O vs T relationship and the d18O vs humidity relationship (i.e. do T and humidity correlate)?

In the abstract I prefer to focus on the temperature vs isotope relationship as it is what the community usually uses.

[...] we find that the temperature vs δ18O relationship is dependent on synoptic events dynamics in winter contrary to summer.

Line 17: Instead of giving an outlook can you specify how this link between vapor and

precipitation helps to interpret short firn cores?

Then, as a clear link is found between the isotopic composition of water vapor and precipitation, we assess how isotopic models can help interpret short firn cores. In fact, a virtual firn core built from ECHAM-wiso outputs explains much more of the variability observed in S1C1 isotopic record than a virtual firn core built from temperature only. Yet, deposition and post-deposition effects strongly affects the firn isotopic signal and probably account for most of the remaining misfits between archived firn signal and virtual firn core based on atmospheric modeling.

Introduction

Line 34: Remove "providing".

done

Line 51: Remove "record".

done

Line 53: I think there is no need to start a new paragraph here.

done

Line 69: In the named study, comparison of water vapor isotope data and back trajectory simulations was performed to … ?

Also, a two-year data series at Neumayer station III was used to associate isotopic signal in water vapor to air masses origin through back trajectory simulations analyses (Bagheri Dastgerdi et al., 2020).

Line 71: suggested change to "the instrumental setup and the 2-year isotopic series". Remove "analysis of a".

Here we present a long-term study of continuous isotopic measurement of water vapor and precipitation at Dumont d'Urville (DDU). First, we present our instrumental set-up and the 2-year isotopic series observed at DDU.

Line 72: Suggested change to "Then, we show the ECHAM6-wiso outputs […]"

Then, we show the ECHAM6-wiso output at DDU geographical position […]

Line 73: Remove "evaluated".

[…] and evaluate the model performance.

Methods

Line 85: The start of this sentence sound strange to me. I suggest to avoid "in the following study we discuss" and reformulating to be more concise, e.g. "The laser spectrometer measures molecular water vapor mixing ratio and …"

The laser spectrometer measures molecular water vapor mixing ratio (in ppmv), […]

Line 92: To be more concise, I suggest: "The d18O and dD series were calibrated following…"

The $\delta 18O$ and $\delta D$ series were calibrated following the approach described […]

Line 102: Please specify "isotope-humidity relationship". You may also highlight that you do not observe a difference in this relationship in the field and at LSCE.

To determine the isotope-humidity relationship for calibration, vapor with known isotopic composition was generated at different humidity levels, from 150 to 1500 ppmv with the low humidity level generator (in the field) and from 1000 to 5500 ppmv with the SDM (at LSCE). A well constrained relationship is determined from the 2018 data set over the whole range of humidity values (Fig. S2). We note that we do not observe differences between field and LSCE relationships.

Line 116: Can you specify how many replicates per sample? (2-3?). You do not use an independent standard that you measure routinely in each sequence that you could use to estimate uncertainty?

The uncertainty (1 sigma) of our data set is 0.2 ‰ and 0.7 ‰ respectively for δ18O and δD. It is estimated using replicates (2 measurements) over 15 % of the samples.

Line 123-125: Can you specify for what these data are used? (Comparison to isotope data? /Identification of processes driving isotope variability of atm water vapor/precipitation / input for ECHAM6-wiso model). If you use the ERA5 reanalysis data solely for nudging the ECHAM6-wiso model, you may consider combining this section with the following.

We have deleted this section in the new version of the manuscript. Indeed, it was solely used for nudging ECHAM6-wiso.

Results

Table 1: Specify that the isotope composition of atmospheric water vapor is shown, not for precipitation. Verify throughout the paper if there is a need to specify if you mean atm water vapor or precipitation.

[…] isotopic composition of water vapour […]

Table 1: Why there is no value for the slope between d18O vs Temp in DJF? I think something shifted in this table (cf. Fig. S4). Please check.

There is no value for the slope between d18O vs Temp in DJF because R² < 0.1 so it would not be representative.

Line 147: According to Table 1, also d-excess shows higher values in summer (10.2) than in winter (7.8), but the difference is not significant?

The difference of d-excess mean value between winter and summer (2.4 ‰) is not significant compared to the standard deviation (3.2 and 3.3 ‰ for DJF and JJA, respectively).

Line 153: "[…] lead to higher variability […] when there are larger meridional temperature and moisture gradients than in summer."

This paragraph has been fully re-written to assess previous comments:

The variability of temperature, δ18O and humidity also shows a seasonality (Tab. 1), with higher standard deviation in winter than in summer. These results can be explained by a different seasonal impact of the main precipitation events. Blue bars in

Figure 2 show the distribution of the 3-day periods centered on daily precipitation rates higher than 4.5 kgm−2 day−1. We detect 35 precipitation peaks over the 2-year period, 7 (20 %) during DJF (summer) and 8 (23 %) during JJA (winter), so no seasonality of the occurrence is observed. Though, the temperature anomalies associated with those events are more important in winter (Fig. S4). Servettaz et al. (2020) demonstrated the key role that such precipitation events, controlled by synoptic scale dynamics, could play in ice-core interpretation at high accumulation sites. As they are associated with warm and moist air intrusions, they cause warm anomalies compared to the seasonal mean. Here, we point out two major events that are particularly intense during extended winter (from May to September): (a) 23 July 2019, with a precipitation rate of 31 kgm−2 day−1 and (b) 2 July 2020, with a precipitation rate of 21 kgm−2 day−1. These events correspond to the largest daily precipitation rates of each winter, and to the first and third largest daily precipitation rates when considering the whole 2019-2020 period. The values of temperature, humidity and δ18O during these winter events (-1 ◦C and -4.4 ◦C, 5780 ppmv and 4370 ppmv, -17.8‰ and -19.3 ‰, respectively for the two events) are close or above summer averages (Tab. 1).

Line 167: I can't find these numbers in Table 1. Please check if something shifted. You may also consider showing the plots for the full period in Fig S3 and S4. Also, I don't think that a slope of 0.5 ‰ºC−1 is significantly different from 0.6 ‰ºC−1. In contrast, the slope seems to be significantly higher in fall and winter than in spring and summer. Please verify.

I have re-written this paragraph. Also, the Table 1 is not missing values; spring and autumn slopes values can be find in Fig S3 and Fig S4.

First, humidity and δ18O show high correlation coefficients (calculated from daily means) both over the whole record (R2=0.6) and at a seasonal scale (R2=0.5 for DJF and JJA). The slope of this linear relationship (Tab. 1 and Fig. S5) is almost doubled during winter (4.5.10−3 ‰ ppmv−1) compared to summer (2.4.10−3 ‰ ppmv−1). This difference between low and high humidity regimes (during winter and summer, respectively) is expected from the relationships between δ18O and humidity along distillation line or during mixing of 2 different air masses (Steen-Larsen et al., 2017). Second, the linear relationship between δ18O and temperature is strong for the full period (R2=0.5) but vanishes during summer (R2<0.1). This can be related to the smaller daily variability during summer, in comparison to winter when synoptic event occurrences lead to larger increase of temperature and δ18O over synoptic time scale. The δ18O-temperature slope (Tab. 1 and Fig. S6) over the full period (0.5 ‰ ◦C−1) is similar to the winter mean slope (0.6 ‰ ◦C−1, R2=0.4). We note that spring mean slope is slightly different (0.4 ‰ ◦C−1, R2=0.3) but is statistically less representative in comparison.

Line 169: Is it the synoptic events that are not visible or rather the impact of the synoptic events on the meteo data?

I have re-written all the paragraph discussing this to assess previous comments.

The variability of temperature, δ18O and humidity also shows a seasonality (Tab. 1), with higher standard deviation in winter than in summer. These results can be explained by a different seasonal impact of the main precipitation events. Blue

155 bars in Figure 2 show the distribution of the 3-day periods centered on daily precipitation rates higher than 4.5 kgm−2 day−1. We detect 35 precipitation peaks over the 2-year period, 7 (20 %) during DJF (summer) and 8 (23 %) during JJA (winter), so no seasonality of the occurrence is observed. Though, the temperature anomalies associated with those events are more important in winter (Fig. S4). Servettaz et al. (2020) demonstrated the key role that such precipitation events, controlled by synoptic scale dynamics, could play in ice-core interpretation at high accumulation sites. As they are associated with warm and moist air intrusions, they cause warm anomalies compared to the seasonal mean.

Line 170: The observation of diurnal cycle only in summer is very interesting. However, std given in Table 1 is based on daily means. Hence, diurnal variability should not affect this value, shouldn't it?

I removed the reference to Table 1.

As mentioned above, synoptic events are not clearly visible in summer. Summer variability is actually dominated by the succession of diurnal cycles. In Figure 3, we show the mean diurnal cycles in summer and winter. During winter, the diurnal cycles of temperature and humidity are flattened to 0.6 ∘C and 40 ppmv, and are not visible for δ18O and d-excess (Fig. 3).

Line 202: Remove "temperature at DDU".

Done

Line 237: You may consider highlighting the two precipitation peak events in figure 5.

Done

Line 238: "The amplitude of these peaks in ECHAM is comparable to that of measurements."

Done

Table 3: Consider combining with Table 1, as half of the presented information are identical.

I agree that half of the information presented is identical in Table 3 and Table 1, but for a better reading experience, I think they should be separated.

Discussion

Line 253: Suggested change to: "In the following, we explore how […]"

Done

Line 254: Suggested change to: "For this, we use […]"

Done

Conclusion

Line 319: I do not see the nice agreement (in absolute values) for all three parameters simultaneously. Approach c fits d18OV, but produces an offset of 400 ppmv in humidity, while temperature is always about 4-5ºC underestimated by the model. Irrespective of the offset in absolute values, however, the observed trends are well reproduced by the model. You may want to specify this here. Also, you do not show model results for approach a and b, so that the reader remains unknown if these model approaches also allow to reproduce general trends. You may do so in the supplement.

I have changed the sentence according to this comment.

We evaluate the ECHAM6-wiso model through a comparison of the simulated δ18O of water vapor and precipitation with our record and we show that a combination of continental (79 %) and oceanic (21 %) grid cells leads the modeled temperature, humidity and δ18O to nicely fit trends and variability of observations.

The model results for approaches a and b are described quantitatively in Table 2 and description in section 3.2. Though, I consider that readers have information to understand combination a and b also improve fits of trend and variability.

Line 326: Typo. "Our results", not "are results".

Done

**Reviewer #2:**

1) **The abstract is a bit cursory without concision, and lacks some important conclusions. I would suggest the authors rewrite it.**

We agree with the reviewer #2 that the abstract could improve significantly. We will rewrite it and add the following conclusions: 1) ECHAM6 wiso is able to reproduce isotopic signal at DDU at different time scale (synoptic and seasonal scale) 2) The low frequency variability is still underestimated in ECHAM6-wiso (wiso model is not expressing enough variability at interannual to decadal scales) and 3) Deposition and post deposition effects contribute significantly to the isotopic signal recorded in coastal site.

Here is the new version of the abstract:

In a context of global warming and sea level rise acceleration, it is key to estimate the evolution of the atmospheric hydrological cycle and temperature in polar regions, which directly influence the surface mass balance of the Arctic and Antarctic ice sheets. Direct observations are available from satellite data for the last 40 years and a few weather data since the 50's in Antarctica. One of the best ways to access longer records is to use climate proxies in firn or ice cores. The water isotopic composition in these cores is widely used to reconstruct past temperature variations. We need to progress in our understanding of the influence of the atmospheric hydrological cycle on the water isotopic composition of ice-core. First, we present a 2-year long time series of vapor and precipitation isotopic composition measurement

at Dumont d'Urville station, in Adelie Land. We characterize diurnal variations of meteorological parameters (temperature, atmospheric water mixing ratio (hereafter humidity) and δ18O) for the different seasons and determine the evolution of key relationships (δ18O versus temperature or humidity) along the year: we find that the temperature vs δ18O relationship is dependent on synoptic events dynamics in winter contrary to summer. Then, this data set is used to evaluate the Atmospheric General Circulation Model ECHAM6-wiso (model version with embedded water stable isotopes) in a coastal region of Adelie Land where local conditions are controlled by strong katabatic winds which directly impact the isotopic signal.We show that a combination of continental (79%) and oceanic (21%) grid cells leads model outputs (temperature, humidity and δ18O) to nicely fit the observations, at different time scales (i.e. seasonal to synoptic). Therefore we demonstrate the added value of long-term water vapor isotopic composition records for model evaluation. Then, as a clear link is found between the isotopic composition of water vapor and precipitation, we assess how isotopic models can help interpret short firn cores. In fact, a virtual firn core built from ECHAM-wiso outputs explains much more of the variability observed in S1C1 isotopic record than a virtual firn core built from temperature only. Yet, deposition and post-deposition effects strongly affects the firn isotopic signal and probably account for most of the remaining misfits between archived firn signal and virtual firn core based on atmospheric modeling.

2) **The description of the isotopic calibration is lacking detail and uncertainties. The plots of drift correction should be provided in SI. Delta 18O-humidity calibration data from SDM are missed in Fig. S2.**

Indeed, there's a problem on figure S2, the data made with SDM are plotted but they are not plotted in the right color (dark gray circles), which explains the confusion, Figures have been replotted and replaced in the Supplementary material.

Regarding the drift correction, in the next manuscript version (in supplementary material) we will add the Figure displayed below showing drift estimation through 48h period routine automatic measurements in comparison with the drift estimated through 4 different humidity sessions. Also, we will comment those results in a paragraph (in the supplementary online material) in order to explain why we do not consider drift correction in regard to the humidity correction applied to the data series.

This explanation is given below and added to the manuscript supplement:

[Figure]

**Figure S3:** Mean linear drift estimation from different sets of data. Black crosses are routine measurements and red crosses are measurements made during humidity calibration sessions. Gray bars are the standard deviation associated with each measurement. Black (red) line is the linear drift estimated from routine measurements (humidity calibration sessions measurement). The green line is the linear drift estimated from both data series

**Text S1:**

To assess the drift of the instrument, standard measurements (black crosses in Figure S3) was performed every 2 days with the humidity generator set at 1100ppmv during 40 minutes (1140ppmv measured on average). Some technical issues led us to select only 150 calibrations over the 2-year period. The results show a drift with decreasing $\delta^{18}O$ and $\delta D$ values with time. Unfortunately, the data are very scattered and even sparse after the first year of installation of the instrument. The reason for this scattering is a problem with the humidity generator (bad drying procedure in the instrument) when it was working without human intervention. However, we could perform proper calibrations each year during the field summer seasons (red crosses). Because we are more confident with these measurements, we have only kept these series for the drift estimation.

Mean drift over the two-year period is hence estimated to 0.01 ‰/years and 0.6 ‰/years for $\delta^{18}O$ and $\delta D$, respectively. Because the drift is very small but associated

with a high uncertainty, we decided not to correct our data series for mean annual drift but to associate a large uncertainty with $\delta^{18}O$ and $\delta D$. The latter is calculated as the 70th percentile of the distribution of the 4 annual calibration during the summer season and results in 0.8 ‰ and 3.2‰, respectively. The new version of the LHLG (low-humidity level generator), installed in January 2022 at DDU, does not show any longer such a scatter in the routine drift calibrations; there is a good agreement between the drift inferred from this routine calibration and the drift calculated from the calibrations performed during the summer season showed in this study.

In the main text we have modified the section referring to data calibration:

The same shift in $\delta 18O$ and $\delta D$ has been observed between measured and true value for both NEEM and FP5. We checked the mean drift of the instrument by measuring NEEM standard at 1100 ppmv using an automatic routine every 48 hours. Some technical issues led us to select only 150 calibrations over the 2-year period (Fig. S3). A large scattering was observed which was due to problem with the humidity generator during the winter and we only used the data acquired during summer field season for the drift calibration (Text S1). The estimated correction associated with the mean linear drift is insignificant with respect to 115 the humidity dependency correction and we estimate the mean uncertainty as 0.8 ‰ and 3.2 ‰ for $\delta 18O$ and $\delta D$ respectively (details in Text S1).

3) **The influence of synoptic events does not address very well. Some descriptions are not clear at seasonal or events scales. For instance, what is the difference of such events between winter and summer, and what is the reason for those distinct influences in L150-155? How could get the conclusion in L315-317?**

Thanks for this comment which echoes that of reviewer #1. Therefore, see my reply to this last:

It seems that our definition and detection of synoptic events is, indeed, unclear and this makes our result and interpretation confusing. Here we are interested in main precipitation events, as in definitive, they are responsible for isotopic signals recorded in firn cores. First, we identified those main precipitation events (using a daily precipitation rate as described in the manuscript) and then we conclude that they must be related to warm and moist air masses intrusion coming from North (as they are really often associated to humidity and temperature increase on a daily time scale), corresponding to synoptic scale events. Then, we observe that occurrences of such events aren't seasonally dependent (DJF vs JJA comparison) but their impact on local weather or climatology is. Indeed, as pointed by reviewer #1, the anomalies (temperature, hum, d18) caused by those intrusions are more important in winter and could significantly impact winter variability, as identified by (Servettaz et al., 2020) at ABN.

In the manuscript changes have been made in the section 3.1. Re-writing of the paragraph introducing events associated with synoptic scale dynamics:

The variability of temperature, $\delta 18O$ and humidity also shows a seasonality (Tab. 1), with higher standard deviation in winter than in summer. These results can be

explained by a different seasonal impact of the main precipitation events. Blue bars in Figure 2 show the distribution of the 3-day periods centered on daily precipitation rates higher than 4.5 kgm−2 day−1. We detect 35 precipitation peaks over the 2-year period, 7 (20 %) during DJF (summer) and 8 (23 %) during JJA (winter), so no seasonality of the occurrence is observed. Though, the temperature anomalies associated with those events are more important in winter (Fig. S4). Servettaz et al. (2020) demonstrated the key role that such precipitation events, controlled by synoptic scale dynamics, could play in ice-core interpretation at high accumulation sites. As they are associated with warm and moist air intrusions, they cause warm anomalies compared to the seasonal mean. Here, we point out two major events that are particularly intense during extended winter (from May to September): (a) 23 July 2019, with a precipitation rate of 31 kgm−2 day−1 and (b) 2 July 2020, with a precipitation rate of 21 kgm−2 day−1. These events correspond to the largest daily precipitation rates of each winter, and to the first and third largest daily precipitation rates when considering the whole 2019-2020 period. The values of temperature, humidity and δ18O during these winter events (-1 ◦C and -4.4 ◦C, 5780 ppmv and 4370 ppmv, -17.8‰ and -19.3 ‰, respectively for 165 the two events) are close or above summer averages (Tab. 1).

As for conclusion, it is not impacted by this last clarification, in addition to the comment #5. We insist on the various impact of synoptic or seasonal time scale variations regarding the type of record we look at: "The warm and wet synoptic events occurring in winter and associated with strong precipitation are clearly imprinted in the water vapor isotopic signal while our snow samplings mainly capture the strong seasonal cycle."

4) **The evaluation of d-excess from ECHAM6-wiso is missing in section 3.2. The d-excess variability is well established from observations, but does not show any related analysis to combine with simulations. Why?**

I thank the reviewer for this comment. Indeed, there is a lack of comment on the comparison between d-excess measured and issued from the ECHAM6-wiso model even though comparison is made on Table 3 and Figure S7. From those data, we observe that ECHAM6-wiso is not able to reproduce the second-order parameter d-excess. In fact, a part of the comparable mean values over the two-year data series (8.4 ‰ and 7.8 ‰ respectively for measurement and model output), model fails in reproducing the seasonality of observed d-excess. Seasonality is actually inverted in the model: while d-excess reached its maximum in summer (JJA) in observations (10.2‰), its value is minimal in ECHAM6-wiso combination (6.5‰).

We have added this short analysis to the revised manuscript in section 3.2.

As for d-excess, while modeled average value (7.8‰) and variability (4.4‰) are close to measured ones (Fig. S9), model fails to reproduce the seasonality of observation. Seasonality is actually inverted in the model: while d-excess reached its maximum in summer (DJF) in observations (10.2‰), its value is minimal in ECHAM6-wiso combination (6.5‰). We will thus only consider the δ18O in the following.

**5) I can not get the point clearly in L243-246. It seems controversial with the conclusion.**

"In particular, the seasonal cycle is well captured both by the observations and model outputs with lower mean δ18O values during winter and higher mean δ18O during summer in both modeled and measured precipitation (Fig. S9). The daily precipitation δ18O samples are however strongly scattered and it is not possible to observe in the precipitation δ18O record (hereafter, δ18Op) an equivalent to the strong peaks observed in the water vapor δ18O during the two strong mid-winter synoptic events (Fig. S9)."

We understand that this paragraph was not clear enough. We have added the following explanations to the end of section 3.1.

Furthermore the daily precipitation δ18O samples are strongly scattered and it is not possible to observe in the precipitation δ18O record an equivalent to the strong peaks observed in the water vapor δ18O during the two strong mid-winter synoptic events. Because the sampling of precipitation was limited to one sample per day and only for the days with precipitation, it is expected that we cannot observe the same δ18O signal in the precipitation record than in the continuous water vapour at an hourly resolution.

Moreover, we do believe that this paragraph is not contradictory with what is written in the conclusion: "The warm and wet synoptic events occurring in winter and associated with strong precipitation are clearly imprinted in the water vapor isotopic signal while our precipitation water isotopic signal only captures the strong seasonal cycle."

**1) Increase all font sizes in figures and figures are too small to see details.**

I have resized every figure of the manuscript.

**2) It is hard to compare the vapor data and precipitation data when they are plotted on separate panels with distinct axis scales in Fig. S9.**

Indeed, the figure S9 does not permit us to make this comparison easily. While comparing signal imprinted in vapor in comparison to precipitation, I should refer to Figure S12. This figure is a scatter plot comparing isotopic signal in vapor versus isotopic signal in precipitation for both model and observation. I will change the referencing in the modified manuscript.